# OpenReview forum: "CAMEL: Confidence-Gated Reflection for Reward Modeling"
_ICML.cc/2026/Conference — ICML 2026 regular_

### Official Review · Reviewer_jTNp · 2026-03-11

**Soundness:** 2
**Presentation:** 3
**Significance:** 3
**Originality:** 2
**Overall Recommendation:** 5
**Confidence:** 3

**Summary:**

The paper proposes CAMEL (confidence-gated reflection for reward modeling). It introduces a two-stage framework that bridges
scalar and generative paradigms through selective reflection. Experiments first empirically demonstrate that confidence scores in single-token predictions are highly correlated with final accuracy. Therefore, the paper proposes a two-stage pipeline, first performing lightweight single-token decisions, and if confidence is insufficient, generating brief reflections before final judgment. In the experiment results, they show that CAMEL-Reflection improves upon CAMEL-fast by enabling more thinking, and CAMEL  shows good accuracy-cost Pareto frontier.

**Compliance With Llm Reviewing Policy:**

Affirmed.

**Ethical Review Concerns:**

No ethical concerns.

**Final Justification:**

I increased my score after reading the author response.

**Key Questions For Authors:**

- Do you think CAMEL training would hurt performance compared to normal judge training?
- Do you expect the confidence-accuracy correlation  to hold for models of different sizes and families? Have you conducted any preliminary experiments with other base models?
- For practitioners deploying CAMEL, what guidance can you provide for setting τ? Is there a validation-based approach, or does the optimal threshold vary significantly across domains?
- In Figure 6, there are cases where reflection changes correct predictions to incorrect ones. What characterizes these failure cases?
- Figure 5 shows the confidence distribution shifts leftward after CAMEL training, do you think it could lead to over-triggering of reflection at inference time?

**Limitations:**

No, the paper does not include an explicit limitations section. Please see weaknesses and questions above for further improvement.

**Strengths And Weaknesses:**

## Soundness
### Strengths:
- The core observation that confidence scores correlate with accuracy is well-motivated and empirically validated across multiple datasets (Figure 2, Figure 7). This provides a principled basis for the gating mechanism.
- The counterfactual prefix augmentation is a clever technique that addresses a real problem: without it, the model might simply echo its initial verdict rather than genuinely reflecting.
### Weaknesses:
- Missing critical baseline: The paper lacks comparison against a natural baseline—standard SFT+RL training where the model generates reasoning before the verdict rather than the reflection structure where an initial verdict exists.
- Limited model diversity: Only Qwen3-14B is examined. The generalizability of the CAMEL across different model families and scales remains unverified.
- Threshold sensitivity: The choice of τ=5 as the default threshold appears somewhat arbitrary. While Figure 4 shows the Pareto frontier, there's no principled method for selecting τ in new deployment scenarios.

## Presentation
The paper is well-written and easy to follow. However, the paper would benefit from:
- A more detailed analysis of what the model learns during reflection (qualitative examples)
- Discussion of failure cases where reflection hurts rather than helps (the 51 cases in Figure 6a)

## Significance
The work addresses a practical problem which is balancing accuracy and efficiency in reward modeling. The approach would be particularly valuable in resource-constrained deployment scenarios.

## Originality
This paper integrates multiple approaches to form an effective combined solution: (1) using token-level confidence as a gating signal, (2) counterfactual prefix augmentation for RL training

---

> ### Author Rebuttal · Authors · 2026-03-31
>
> We sincerely thank the reviewer for the detailed feedback.
>
> > **W1:** Missing critical baseline — standard SFT+RL training where the model generates reasoning before the verdict, rather than CAMEL's verdict-first reflection structure.
>
> Thank you for this helpful suggestion. We ran the requested controlled experiment using the same data, SFT+RL recipe, and base model (Qwen3-14B), but with the standard "reasoning → verdict" structure instead of CAMEL's verdict-first design.
>
> | Model              | RewardBench | RM-Bench | JudgeBench | Avg. |
> | ------------------ | ----------- | -------- | ---------- | ---- |
> | Qwen3-14B-SFT-GRPO | 91.1        | 72.9     | 66.0       | 76.7 |
> | CAMEL              | 92.4        | 81.9     | 69.1       | 81.1 |
> | CAMEL-Reflection   | 92.8        | 84.2     | 71.6       | 82.9 |
>
> This comparison shows that the conventional reasoning-before-verdict baseline remains substantially below both CAMEL and CAMEL-Reflection (**76.7% vs. 81.1% / 82.9%**). We believe this directly supports improvement stems from CAMEL's training design (verdict-first + counterfactual augmentation), not merely standard RL training.
>
> > **W2:** Generalizability across different model families and scales remains unverified.
>
> We appreciate this concern. We validate the confidence signal across **six model families** (8B–32B) on five datasets and train a Qwen3-32B CAMEL variant. Due to space constraints, we present the full results in our response to Reviewer 87N2 (W2 for multi-model study, W3 for 32B scaling). In brief, CAMEL scales effectively and the confidence signal generalizes across families.
>
> > **W3:** No principled method for selecting τ in new deployment scenarios.
>
> Please see Reviewer 1hc8 (W1) for detailed analysis of confidence and how to choose threshold  $\tau$. Briefly, $\tau$ controls a **continuous trade-off** on the Pareto frontier (Figure 4): lower $\tau$ triggers more reflection, while higher $\tau$ saves tokens. In our experiments, $\tau \in [3, 7]$ **consistently works well** across all benchmarks.
>
> > **Presentation:** (1) A more detailed analysis of What reflection learns. (2) Failure cases where reflection hurts.
>
> **(1) What reflection learns.** Three recurring patterns:
>
> - **criteria re-evaluation** — re-examining both responses following the criteria hierarchy (safety → accuracy → relevance → comprehensiveness);
>
> - **fine-grained comparison** — identifying specific differences rather than holistic judgment;
>
> - **verdict override** — revising when analysis contradicts the initial snap judgment.
>
> We will include examples of these patterns in the revised appendix.
>
> **(2) Failure cases.** We analyze all hurt cases (initially correct $\rightarrow$ incorrect after reflection):
>
> | Benchmark   | Hurt Cases | Unknown Verdict | Fraction |
> | ----------- | ---------- | --------------- | -------- |
> | RewardBench | 51         | 16              | 31.4%    |
> | RM-Bench    | 332        | 131             | 39.5%    |
> | JudgeBench  | 72         | 49              | 68.1%    |
>
> The dominant failure mode is generation non-convergence: the reflection becomes too long and fails to produce a parseable final verdict ("Unknown"), rather than confidently reaching the wrong conclusion. Among genuine reasoning failures, hurt cases concentrate in safety close-calls and reasoning-heavy domains such as math and code. We will include representative examples in the revised appendix.
>
> > **Q1:** Would CAMEL training hurt performance compared to normal judge training?
>
> No. As shown in W1, **CAMEL-Reflection (82.9%)** substantially outperforms the standard SFT+GRPO judge (**76.7%**).
>
> > **Q2:** Do you expect the confidence-accuracy correlation to hold for models of different sizes and families? Have you tested with other base models?
>
> Yes. We validate this across six model families (8B-32B) on five datasets, and the positive correlation holds consistently. Under the space limit, we respectfully refer the reviewer to our response to Reviewer 87N2 (W2) for full results.
>
> > **Q3:** For practitioners deploying CAMEL, what guidance can you provide for setting τ?
>
> Please see W3 above and Reviewer 1hc8 (W1). In short, $\tau \in [3, 7]$ is stable across benchmarks, and a held-out validation set or deployment budget is the natural way to choose it.
>
> > **Q4:** In Figure 6, what characterizes the cases where reflection changes correct predictions to incorrect ones?
>
> See the failure case analysis under "Presentation" above.
>
> > **Q5:** Could confidence distribution shift lead to over-triggering of reflection at inference time?
>
> We believe the leftward shift is actually desirable: after CAMEL training, the model becomes more conservative on uncertain cases to trigger reflection and improve accuracy. Even if fully over-triggered (CAMEL-Reflection), the cost is only comparable to standard GenRMs — but with **higher accuracy**. As noted in W3, selecting $\tau \in [3, 7]$ effectively controls the reflection rate.

---

> > ### Author Rebuttal · Reviewer_jTNp · 2026-04-04
> >
> > Thanks for the author response, my concerns are resolved.

---

> > > ### Author Response · Authors · 2026-04-04
> > >
> > > We appreciate the reviewer’s careful reconsideration and are pleased that our response has resolved the concerns. We will further polish the paper by incorporating the reviewer’s helpful suggestions through minor clarifications and refinements.

---

### Official Review · Reviewer_1hc8 · 2026-03-11

**Soundness:** 2
**Presentation:** 3
**Significance:** 3
**Originality:** 2
**Overall Recommendation:** 4
**Confidence:** 3

**Summary:**

The paper proposes CAMEL, a confidence-gated reflection method for reward modeling that combines the benefits of scalar RMs and generative RMs. The authors use the initial verdict token's log-probability margin as a confidence score to decide whether to trigger a reflection before the final verdict token. Then they use GRPO with counterfactual prefix augmentation that forces the model to diverse initial verdicts and encourages genuine revision. CAMEL is evaluated on multiple RM evaluation benchmarks such as RewardBench, RM-Bench, and JudgeBench.

**Compliance With Llm Reviewing Policy:**

Affirmed.

**Final Justification:**

The author rebuttal addressed most of my concerns. The remaining issues, such as the discussion of overconfidence samples and the threshold can be addressed through revision to the paper.

**Key Questions For Authors:**

* Could the authors add an analysis in what fraction of high-confidence predictions are wrong?
* How is the confidence threshold selected across datasets/domains?
* How sensitive is CAMEL to the base model? Would the performance scale with smaller/larger models?

**Limitations:**

yes

**Strengths And Weaknesses:**

## Strengths
* The problem to bridge scalar and generative RMs is well-motivated. The method is simple and intuitive.
* The evaluation covers broad RM benchmarks.
* The authors provide some useful analyses, such as the confidence-accuracy relationship and ablation studies.

## Weaknesses
* Although the confidence gating is intuitive, it is not necessarily calibrated across domains or models. In my opinion, the authors should conduct a further calibration study (ECE or ROC for confidence-as-difficulty). There is no principled way to choose the threshold for different models. There is limited analysis of failure modes when high-confidence errors bypass reflection.
* The method is highly based on entropy/confidence-based paradigm. However, the adaptive-compute or confidence-gated methods, such as [1, 2, 3], are not cited or discussed. The authors should clarify the novelty and distinction from prior works.
* Generalization to BoN selection (which is crucial for practical RM) is not evaluated.

[1] Taubenfeld, Amir, et al. "Confidence improves self-consistency in llms." Findings of the Association for Computational Linguistics: ACL 2025. 2025.

[2] Fu, Yichao, et al. "Deep think with confidence." arXiv preprint arXiv:2508.15260 (2025).

[3] Sharma, Aman, and Paras Chopra. "Think just enough: Sequence-level entropy as a confidence signal for llm reasoning." arXiv preprint arXiv:2510.08146 (2025).

---

> ### Author Rebuttal · Authors · 2026-03-31
>
> We sincerely thank the reviewer for the thorough feedback and valuable suggestions.
>
> > **W1:** The confidence gating is not necessarily calibrated across domains or models. A calibration study (ECE or AUROC) is needed. There is no principled way to choose the threshold. Limited analysis of failure modes when high-confidence errors bypass reflection.
>
> We appreciate this suggestion and address all three in turn.
>
> **(1) Calibration study.** We compute AUROC treating $c(x)$ as a binary discriminator of prediction correctness across **six model families** and **five datasets**:
>
> | Model                 | Skywork-80K | Code-Pref | Math-Step | RewardBench | RM-Bench | Avg.  |
> | --------------------- | ----------- | --------- | --------- | ----------- | -------- | ----- |
> | Qwen3-32B             | 0.625       | 0.951     | 0.792     | 0.776       | 0.695    | 0.768 |
> | Llama-3.1-8B-Instruct | 0.554       | 0.804     | 0.575     | 0.692       | 0.653    | 0.656 |
> | GLM-4-9B-Chat         | 0.548       | 0.629     | 0.562     | 0.663       | 0.627    | 0.606 |
> | Phi-4                 | 0.655       | 0.925     | 0.736     | 0.767       | 0.724    | 0.761 |
> | Gemma-3-12B-IT        | 0.577       | 0.800     | 0.705     | 0.698       | 0.676    | 0.691 |
> | Mistral-Small-24B     | 0.675       | 0.920     | 0.714     | 0.750       | 0.699    | 0.752 |
>
> All average AUROC values range from 0.61 to 0.77, confirming the confidence signal **reliably discriminates correct from incorrect predictions across diverse models and domains**.
>
> **(2) Threshold selection.** $\tau$ controls a **continuous Pareto frontier** (Figure 4): lower $\tau$ routes more samples to reflection and improves accuracy at higher token cost; higher $\tau$ skips reflection more aggressively and saves tokens with some accuracy trade-off. There is no single universally "correct" threshold; practitioners choose an operating point according to their deployment budget. In our experiments, $\tau \in [3, 7]$ consistently achieves a good balance across benchmarks.
>
> **(3) High-confidence failure analysis.** We report the error rate among predictions that bypass reflection (confidence $>= \tau$) at $\tau = 5$:
>
> | Benchmark   | High-Conf Samples | Errors | Error Rate |
> | ----------- | ----------------- | ------ | ---------- |
> | RewardBench | 2008 (67.3%)      | 61     | 3.0%       |
> | RM-Bench    | 7007 (58.7%)      | 877    | 12.5%      |
> | JudgeBench  | 210 (33.9%)       | 48     | 22.9%      |
>
> On easier benchmarks, the gate is highly reliable (**3% error** on RewardBench). On harder benchmarks, it **automatically becomes more conservative** — only 34% of JudgeBench samples pass, routing the majority to reflection. We manually inspect high-confidence errors and identify three recurring failure patterns: (a) over-refusal/safety bias — the model confidently penalizes a response that appropriately addresses the query; (b) subtle style-comprehensiveness preferences — overconfidence when responses differ only in fine-grained stylistic aspects; (c) long-chain reasoning misjudgment — confidently endorsing an incorrect reasoning chain. We will include examples in the revised appendix.
>
> > **W2:** The method is highly based on confidence/entropy. However, adaptive-compute or confidence-gated methods [1, 2, 3] are not cited or discussed.
>
> We thank the reviewer for these references and will cite and discuss them in the revised paper. All three use confidence/entropy to reduce computation in reasoning tasks: [1] weights majority voting, [2] filters parallel traces, [3] uses sequence-level entropy for early stopping. CAMEL differs in three ways:
>
> - **Task domain** — CAMEL addresses reward modeling (binary preference judgment) with confidence on a fixed two-token label space, structurally distinct from confidence over free-form reasoning.
> - **Usage** — confidence gates a separate reflection stage, not sample filtering.
> - **Training** — unlike the purely inference-time methods in [1–3], CAMEL includes RL training with counterfactual prefix augmentation that teaches the model to reflect and self-correct.
>
> > **W3:** Generalization to BoN selection (crucial for practical RM) is not evaluated.
>
> We add Best-of-N evaluation using RewardBench 2. Due to space constraints, we present full results (including a larger Qwen3-32B variant) in our response to Reviewer 87N2 (W3). In brief, **CAMEL generalizes effectively to the BoN setting**.
>
> > **Q1:** What fraction of high-confidence predictions are wrong?
>
> See the high-confidence error table in W1 above.
>
> > **Q2:** How is the confidence threshold selected across datasets/domains?
>
> See threshold discussion in W1.
>
> > **Q3:** How sensitive is CAMEL to the base model? Would performance scale with smaller/larger models?
>
> We train a larger CAMEL based on Qwen3-32B and observe consistent improvements. Full scaling results are in our response to Reviewer 87N2 (W3), where **CAMEL demonstrates effective scaling to larger backbones**.

---

> > ### Author Rebuttal · Reviewer_1hc8 · 2026-04-03
> >
> > Thank you for your response. It has addressed most of my concerns. I still have some questions regarding the overconfidence samples, since the error rate on the more challenging benchmarks seems to be high. I also have a question about the practicality of this framework, as it depends on a threshold. So how this threshold varies and how it should be chosen for different samples, domains, or benchmarks? Some examples would be helpful.

---

> > > ### Author Response · Authors · 2026-04-04
> > >
> > > We sincerely thank the reviewer for the thoughtful follow-up.
> > >
> > > > **Follow-up Q1:** I still have some questions regarding the overconfidence samples, since the error rate on the more challenging benchmarks seems to be high.
> > >
> > > We agree that high-confidence predictions are not error-free, especially on challenging benchmarks. However, the relevant question for confidence gating is not whether the high-confidence group is perfect, but whether it is **substantially cleaner** than the low-confidence group. At $\tau=5$:
> > >
> > > | Benchmark   | High-Conf Fraction | Error (High-Conf) | Error (Low-Conf) |
> > > | ----------- | ------------------ | ----------------- | ---------------- |
> > > | RewardBench | 67.3%              | 3.0%              | 19.0%            |
> > > | RM-Bench    | 58.7%              | 12.5%             | 37.3%            |
> > > | JudgeBench  | 33.9%              | 22.9%             | 42.9%            |
> > >
> > > Across all three benchmarks, high-confidence predictions have **significantly lower error rates** than low-confidence ones (1.9–6.3$\times$ reduction), confirming that confidence gating reliably separates easier from harder instances regardless of benchmark difficulty.
> > >
> > > > **Follow-up Q2:** I also have a question about the practicality of this framework, as it depends on a threshold. So how this threshold varies and how it should be chosen for different samples, domains, or benchmarks? Some examples would be helpful.
> > >
> > > Thank you for raising this important practical question. We would like to first reframe this concern: $\tau$ is a **core advantage** of CAMEL rather than a tuning burden. It gives practitioners **direct, inference-time control over the accuracy–efficiency trade-off** — a capability that existing GenRM methods lack. Methods such as RM-R1 are locked into a single operating point; moving along the accuracy–cost curve requires retraining with a different backbone (cf. Figure 4). CAMEL traverses this entire frontier at deployment time.
> > >
> > > We further address this concern from three aspects.
> > >
> > > **(1) Practical workflow.** Selecting the threshold $\tau$ follows a simple recipe:
> > >
> > > 1. Run CAMEL-Fast (one-token prediction) on a held-out validation set. This costs **<1% of full GenRM inference** and yields confidence scores for all samples at no additional cost.
> > > 2. Sweep $\tau$ on this set and pick the operating point matching the deployment goal (a target accuracy, or a latency/cost budget).
> > > 3. If no labeled set is available, $\tau=5$ is a robust default; alternatively, one can simply reflect the lowest-confidence fraction under a fixed compute budget.
> > >
> > > **(2) Worst-case analysis.** Even at the two extremes of $\tau$, CAMEL's performance boundaries remain strictly favorable:
> > >
> > > - **$\tau$ too low** ($\tau \to 0$, i.e. CAMEL-Fast): **<1% of GenRM inference cost**, while already matching strong baselines (76.8% avg, on par with LDL-Reward-27B and Nemotron-70B in Table 1).
> > > - **$\tau$ too high** ($\tau \to \infty$, i.e. CAMEL-Reflection): inference cost comparable to (or even smaller than) standard GenRM such as RM-R1 (cf. Figure 4), but **accuracy substantially higher** (82.9% vs. 79.7% best prior).
> > >
> > > Any $\tau$ between these extremes selects a point on a Pareto frontier that strictly dominates existing GenRM baselines (Figure 4). The threshold is therefore **highly forgiving**.
> > >
> > > **(3) Concrete examples from our experiments.** Each cell below reports accuracy / reflection rate / average output tokens with different choices of $\tau$:
> > >
> > > | Benchmark   | $\tau=3$            | $\tau=5$             | $\tau=7$             |
> > > | ----------- | ------------------- | -------------------- | -------------------- |
> > > | RewardBench | 93.5% / 10.2% / 104 | 94.2% / 32.7% / 298  | 94.3% / 74.2% / 560  |
> > > | RM-Bench    | 81.3% / 16.7% / 219 | 84.9% / 41.3% / 505  | 87.2% / 76.1% / 784  |
> > > | JudgeBench  | 65.6% / 34.4% / 610 | 67.1% / 66.1% / 1114 | 69.0% / 89.8% / 1458 |
> > >
> > > The default $\tau=5$ in our paper is simply a convenient mid-range value; as the table shows, any $\tau \in [3,7]$ yields competitive performance across all benchmarks, and the threshold can be chosen coarsely without careful tuning.
> > >
> > > A notable pattern: **at the same $\tau$, harder benchmarks automatically route more samples to reflection** (e.g., at $\tau=3$: RewardBench 10.2% vs. JudgeBench 34.4%). This confirms that the confidence score is inherently domain-adaptive — a single global $\tau$ naturally allocates more reflection where it is needed most, without per-domain tuning.

---

### Official Review · Reviewer_YUfX · 2026-03-12

**Soundness:** 3
**Presentation:** 3
**Significance:** 3
**Originality:** 2
**Overall Recommendation:** 5
**Confidence:** 4

**Summary:**

This research works towards building a better generative reward models. Based on the empirical evidence that the log-probability margin between verdict tokens strongly correlates with prediction accuracy, the authors proposed CAMEL, a new response format and a confidence-gated reflection mechanism. To realize the reflection ability, they train the model using RL with counterfactual prefix augmentation. Empirically, they compared CAMEL with various scalar reward models and generative reward models across three RM benchmarks (e.g., RewardBench, RM-Bench, JudgeBench) with 82.9% average accuracy, with 14B parameters.

**Compliance With Llm Reviewing Policy:**

Affirmed.

**Final Justification:**

Authors have fully addressed my concerns, so l increased from 4 to 5.

**Key Questions For Authors:**

Please see the above weakness points.

**Limitations:**

yes

**Strengths And Weaknesses:**

Strengths:
1. The empirical observation shown in Figure 2 is interesting and solid.
2. The new response format (starting with a verdict token, and then reflect and give another verdict token) and RL training with Counterfactual Prefix Augmentation are novel and technically sound.
3. Paper writing is clear.
4. Empirical evaluation and analysis are comprehensive.

Weakness/questions:
1. Could you explain which prompt and response format is used for Qwen3-14B in Figure 2? Are they same with the CAMEL model, and why Qwen3-14B is not in the benchmark results, e.g., GenRMs category.
2. It would be helpful for audience if the authors could include some comparative examples in the appendix to illustrate the differences in response formats across various generative reward models.
3. It seems that CAMEL skip the judge sequences before the first verdict token, compared with the generative RM formation introduced in Section 2, any reasons here?
4. What is the key difference between CAMEL and RM-R1? It seems that both two used RL to train the model using the final accuracy signal?

---

> ### Author Rebuttal · Authors · 2026-03-31
>
> We sincerely appreciate the reviewer's insightful comments.
>
> > **Q1:** Which prompt and response format is used for Qwen3-14B in Figure 2? Are they the same as CAMEL? Why is Qwen3-14B not listed in the GenRM benchmark results?
>
> Thank you for this opportunity to clarify. For Figure 2, we use the prompt and response format shown in Appendix Figure 8. This prompt asks the model to output a single verdict without explicit reflection. Figure 2 isolates the proposed confidence score from the reflection mechanism.
>
> We do include the original Qwen3-14B as a baseline. To avoid redundancy with the main comparison table, we report its results in the ablation study (Section 4.4, Table 2). In Table 2, "Qwen3-14B" uses the prompt in Figure 8, while "Qwen3-14B + Reflection" uses the CAMEL prompt (Figure 3). Under the same full-reflection setting, CAMEL-Reflection outperforms Qwen3-14B + Reflection by **+9.1%** average accuracy (82.9% vs. 73.8%), isolating the contribution of our training pipeline rather than prompt format alone. Even with selective reflection, CAMEL still achieves **+7.3%** over Qwen3-14B + Reflection.
>
> > **Q2:** It would be helpful to include comparative examples of response formats across various generative reward models.
>
> We appreciate this suggestion and will include detailed comparative examples in the revised appendix. Existing generative reward models broadly fall into three response format categories:
>
> 1. **Standard GenRM format** (e.g., GPT-4o, Claude-3.5, J1, RM-R1): the model first generates a reasoning or judgment sequence, then outputs a final verdict. The main differences lie in how reasoning is guided, e.g., J1 emphasizes chain-of-thought via RL, while RM-R1 emphasizes task-specific rubric generation.
> 2. **CAMEL-Fast**: the model directly outputs a single initial verdict token, with no reasoning or reflection.
> 3. **CAMEL / CAMEL-Reflection**: the model first outputs an initial verdict, then generates a short reflection, and finally outputs a final verdict. This "verdict → reflection → verdict" format is a key structural distinction from prior GenRMs, which follow "reasoning → verdict."
>
> > **Q3:** CAMEL skips the judge sequence before the first verdict token, compared with the generative RM formulation in Section 2. Any reasons?
>
> Thank you for this thoughtful question. Section 2 describes the general generative RM paradigm, where a judgment sequence $J$ precedes the verdict. CAMEL intentionally departs from this ordering for a central design reason.
>
> CAMEL's objective is to: (1) obtain a verdict-token distribution **at minimal cost**, (2) compute a confidence margin, and (3) **invoke reflection only for low-confidence instances**. If we followed the standard GenRM format and generated a full judgment sequence before the first verdict, we would already have paid the reasoning cost on every instance, defeating early termination.
>
> By placing the verdict token first, CAMEL makes a **pay-as-you-need** decision: easy instances terminate immediately, while only uncertain instances proceed to reflection. This enables the accuracy-efficiency Pareto frontier in Section 4.3 (Figure 4).
>
> > **Q4:** What is the key difference between CAMEL and RM-R1? It seems that both use RL to train the model using the final accuracy signal.
>
> We appreciate this question. While both use RL with accuracy-based rewards, the differences are substantial.
>
> **First**, the inference paradigm differs fundamentally. RM-R1 treats reward modeling as reasoning: **every instance undergoes full judgment** regardless of difficulty. CAMEL first produces a single-token verdict and uses the confidence margin to decide whether reflection is needed, yielding a strictly better **accuracy–efficiency Pareto frontier** (Section 4.3, Figure 4).
>
> **Second**, the training objective targets different capabilities. In RM-R1, RL reward is applied to an entire reasoning trace, training better end-to-end reasoning chains. In CAMEL, the initial verdict $v_0$ is fixed context; RL credit is assigned only to the reflection $J$ and final verdict $v_1$. CAMEL's objective is learning **when to confirm and when to revise** — a self-correction capability distinct from "reasoning → answer."
>
> **Third**, **counterfactual prefix augmentation** is unique to CAMEL. For each training instance $(x, z)$, we construct two versions with forced initial verdicts ($v_0 = A$ and $v_0 = B$), while the reward depends only on the final verdict. This forces the reflective stage to learn genuine revision rather than simple imitation, and has no direct counterpart in RM-R1. This distinction is also supported by our ablation in Table 2.
>
> In summary, RM-R1 uses uniform reasoning-style compute, whereas CAMEL bridges scalar and generative reward modeling through **confidence-gated selective reflection**.

---

> > ### Author Rebuttal · Reviewer_YUfX · 2026-04-03
> >
> > Thanks authors to address my concerns. l have increased my score from 4 to 5.

---

> > > ### Author Response · Authors · 2026-04-04
> > >
> > > We sincerely thank the reviewer for their time and reassessment of our paper. We will incorporate the reviewer’s valuable suggestions through minor refinements to further improve the paper.

---

### Official Review · Reviewer_87N2 · 2026-03-13

**Soundness:** 3
**Presentation:** 3
**Significance:** 2
**Originality:** 2
**Overall Recommendation:** 5
**Confidence:** 4

**Summary:**

This paper proposes CAMEL, a confidence-gated reflection framework for reward modeling. The key idea is to use the log-probability margin of the verdict token as a lightweight confidence signal: the model exits early on high-confidence cases and invokes reflection only on uncertain ones.

**Compliance With Llm Reviewing Policy:**

Affirmed.

**Final Justification:**

My concerns have been adequately addressed, and I have raised my score from 4 to 5.

**Key Questions For Authors:**

Please refer to the  Weaknesses.

**Limitations:**

Yes.

**Strengths And Weaknesses:**

Strengths

1. Well-motivated problem: adaptive allocation of reasoning cost for reward modeling is practically important.

2. Simple and clean method: confidence from verdict-token margin is lightweight and easy to deploy.

3. Strong empirical results: CAMEL-Reflection outperforms prior baselines, and gated CAMEL gives a favorable Pareto frontier.

 Weaknesses

1. The method feels somewhat incremental: token-margin confidence + conditional reflection + RL is effective, but the novelty is moderate.

2. The confidence argument is largely empirical/correlational; the paper does not fully explain why this signal should remain robust across model families or shifted settings.

3. Empirical validation is somewhat narrow, mostly centered on the same benchmark family and backbone setting.

---

> ### Author Rebuttal · Authors · 2026-03-31
>
> We sincerely thank the reviewer for the careful reading.
>
> > **W1:** The method feels somewhat incremental: token-margin confidence + conditional reflection + RL is effective, but the novelty is moderate.
>
> We appreciate this comment. CAMEL's contribution lies not in any single component, but in **a principled framework that bridges scalar and generative reward modeling** under an accuracy-efficiency trade-off. Our novelty is threefold.
>
> **First**, we identify a strong empirical correlation between verdict-token log-probability margin and pairwise judging correctness, showing that this zero-cost signal is an effective proxy for instance difficulty.
>
> **Second**, we turn this observation into **confidence-gated selective reflection**: easy cases terminate after one verdict token, while only uncertain cases trigger additional reasoning. This is what creates the accuracy-efficiency trade-off.
>
> **Third**, we introduce **counterfactual prefix augmentation**, which exposes the model to both possible initial verdicts and prevents the reflective stage from merely echoing the first decision.
>
> While the pipeline may appear clean and simple, we view this as a strength — **principled design choices grounded in empirical evidence**, where each component enables the accuracy–efficiency trade-off previously unavailable in reward modeling. We believe that providing novel insights and a well-motivated combination of techniques constitutes genuine contribution, especially when it opens a previously unavailable capability.
>
> > **W2:** The confidence argument is largely empirical/correlational; the paper does not fully explain why this signal should remain robust across model families or shifted settings.
>
> Thank you for raising this important point. We address it from both expanded empirical evidence and a conceptual explanation.
>
> **(1) Extended empirical evidence.** Beyond the Qwen3-14B results in the paper, we evaluate the confidence-accuracy correlation across **six LLM families** and **five datasets** spanning both preference-learning data and evaluation benchmarks. Across all six model families with sizes ranging from 8B to 32B and all five datasets, **the positive correlation remains robust and consistent**. We reproduce Figure 2 for every combination and results are available [here](https://anonymous.4open.science/r/icml-3998-rebuttal/Code-Phi-4.png).
>
> **(2) Conceptual justification.** The confidence signal is grounded in well-studied phenomena. Prior work has established that softmax-based output probabilities carry meaningful confidence information for detecting misclassified examples [1], that pre-trained Transformers exhibit reasonable calibration in classification settings [2], and that LLMs are well-calibrated on structured tasks with constrained output spaces [3].
>
> Crucially, CAMEL's confidence operates on a **fixed binary label space** (A/B verdict) rather than estimating uncertainty over free-form text, inherently more amenable to reliable estimation from logits than sequence-level likelihood over free-form text [2]. The underlying mechanism - probability concentration in constrained binary classification - is **a general property of neural language models**, not a model- or data-specific artifact.
>
> > **W3:** Empirical validation is somewhat narrow, mostly centered on the same benchmark family and backbone setting.
>
> **First**, as described in W2, we validate the confidence-gating signal across **six model families on five diverse datasets**, demonstrating robustness beyond the originally reported setting.
>
> **Second**, we train a **larger CAMEL based on Qwen3-32B**, showing that the framework scales effectively to stronger backbones.
>
> **Third**, we add **RewardBench 2** as an additional Best-of-N evaluation benchmark. Results:
>
> | Model                  | RewardBench | RM-Bench | JudgeBench | RewardBench 2 (BoN) | Avg. |
> | ---------------------- | ----------- | -------- | ---------- | ------------------- | ---- |
> | RM-R1-Qwen-32B         | 89.0        | 73.1     | 64.8       | 64.6                | 72.9 |
> | CAMEL (14B)            | 92.4        | 81.9     | 69.1       | 72.4                | 78.9 |
> | CAMEL-Fast (32B)       | 92.0        | 78.6     | 70.5       | 70.0                | 77.8 |
> | CAMEL (32B)            | 93.5        | 84.3     | 71.1       | 72.6                | 80.4 |
> | CAMEL-Reflection (32B) | 94.1        | 87.6     | 71.6       | 74.6                | 82.0 |
>
> These additions show that CAMEL is not tied to one benchmark family or one backbone size. It scales to 32B and transfers to the practical BoN setting while preserving the same fast / adaptive / full-reflection pattern.
>
> **References**
>
> [1] Hendrycks, D. & Gimpel, K. *A Baseline for Detecting Misclassified and Out-of-Distribution Examples in Neural Networks*.
>
> [2] Desai, S. & Durrett, G. *Calibration of Pre-trained Transformers*.
>
> [3] Kadavath, S. et al. *Language Models (Mostly) Know What They Know*.

---

> > ### Author Rebuttal · Reviewer_87N2 · 2026-04-02
> >
> > My concerns have been adequately addressed, and I have raised my score from 4 to 5.

---

> > > ### Author Response · Authors · 2026-04-02
> > >
> > > We sincerely thank the reviewer for the constructive discussion and for re-evaluating our work. We are happy to address any further questions or comments.

---

### Decision · Program_Chairs · 2026-04-30

**Decision:**

Accept (regular)

**Comment:**

After considering the reviews and the authors’ rebuttal, I recommend accepting this paper. The paper addresses an important problem in reward modeling and proposes a simple, well-motivated confidence-gated reflection framework with empirical evaluation and a favorable accuracy-efficiency trade-off. The main reviewer concerns were about novelty, threshold selection, and generalization, and the rebuttal addressed these issues sufficiently for acceptance. Overall, this is a technically solid and timely contribution, and the remaining limitations are minor and suitable for revision in the final version.